

# Use and misuse of temperature normalisation in meta-analyses of thermal responses of biological traits

Dimitrios - Georgios Kontopoulos[1,2], Bernardo García-Carreras[2], Sofía Sal[2], Thomas P. Smith[2] and Samraat Pawar[2]

[1] Science and Solutions for a Changing Planet DTP, Imperial College London, London, United Kingdom
[2] Department of Life Sciences, Silwood Park, Imperial College London, Ascot, Berkshire, United Kingdom

## ABSTRACT

There is currently unprecedented interest in quantifying variation in thermal physiology among organisms, especially in order to understand and predict the biological impacts of climate change. A key parameter in this quantification of thermal physiology is the performance or value of a rate, across individuals or species, at a common temperature (temperature normalisation). An increasingly popular model for fitting thermal performance curves to data—the Sharpe-Schoolfield equation—can yield strongly inflated estimates of temperature-normalised rate values. These deviations occur whenever a key thermodynamic assumption of the model is violated, i.e., when the enzyme governing the performance of the rate is not fully functional at the chosen reference temperature. Using data on 1,758 thermal performance curves across a wide range of species, we identify the conditions that exacerbate this inflation. We then demonstrate that these biases can compromise tests to detect metabolic cold adaptation, which requires comparison of fitness or rate performance of different species or genotypes at some fixed low temperature. Finally, we suggest alternative methods for obtaining unbiased estimates of temperature-normalised rate values for meta-analyses of thermal performance across species in climate change impact studies.

## INTRODUCTION

Temperature is a key factor that directly or indirectly governs the performance of biochemical reaction rates, physiological rates (e.g., respiration and photosynthesis), and even ecological rates (e.g., prey encounter rate). Understanding how biological rates respond to changes in environmental temperature (the thermal performance curve, TPC; Fig. 1) is important for ecological and comparative evolutionary analyses of thermal physiology, for better predicting how climate change will influence the dynamics of populations, communities, and ecosystems (*Brown et al., 2004*; *Pörtner et al., 2006*; *Dell, Pawar & Savage, 2011*; *Hoffmann & Sgrò, 2011*; *Schulte, Healy & Fangue, 2011*; *Pawar, Dell & Savage, 2015*). Another example of such analyses involves testing the hypothesis of metabolic cold adaptation (MCA; e.g., see *Seibel, Dymowska & Rosenthal, 2007*; *White, Alton & Frappell, 2012*; *Clarke, 2017*), according to which cold-adapted individuals exhibit

Corresponding author
Dimitrios - Georgios Kontopoulos,
d.kontopoulos13@imperial.ac.uk

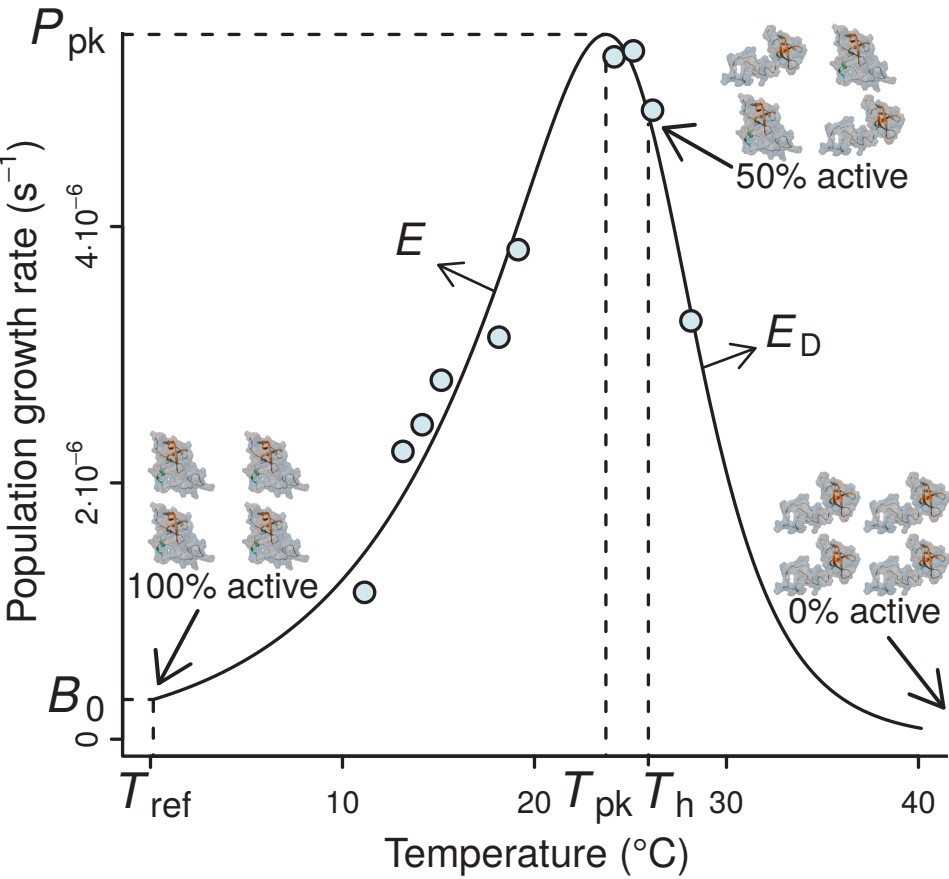

**Figure 1** **A typical example of the four-parameter Sharpe-Schoolfield model fitted to a thermal perfor-mance curve of *Prochlorococcus marinus* strain MIT9515 (*Johnson et al., 2006*).** As depicted, the model assumes that the activity of a single rate-controlling enzyme controls the apparent temperature depen-dence of the rate. $T_h$ is defined as the temperature (before or after the peak) at which 50% of enzyme units are made inactive. Beyond $T_h$, an increasing proportion of the enzyme population is deactivated, to the point where all of them become non-functional, and the curve falls to zero. $B_0$ accurately represents the real rate performance at a reference temperature ($T_{ref}$), only if the enzyme population is fully functional at this particular $T_{ref}$, i.e., $T_{ref} \ll T_h$; otherwise, $B_0$ will necessarily be greater than the real rate value at $T_{ref}$ ($B(T_{ref})$).

higher metabolic rates at low temperatures (well below $T_{pk}$; see Fig. 1) than individuals adapted to higher temperatures.

The TPCs of fundamental biological rates (traits) are generally unimodal, and biological rate versus temperature relationships are typically well-described by mathematical models that quantify four key features of the response: the temperature where the performance peaks ($T_{pk}$), the rate performance at a reference temperature ($B_0$), typically well below $T_{pk}$ within its operational temperature range (*Pawar et al., 2016*), the rise of the rate up to $T_{pk}$ ($E$), and the fall after $T_{pk}$ ($E_D$) (Fig. 1). The normalised rate value $B_0$ is particularly important, as it allows rate performance to be standardised for comparison across individuals and species (*Gillooly et al., 2001*). In particular, the inference of normalised rate values at a reference temperature between species is key for studying MCA, or for

comparisons of the performance of different biological rates (e.g., photosynthesis and respiration) at a common temperature (e.g., see *Padfield et al., 2017*).

Partly mechanistic models that explicitly link a cellular, organismal, or population rate's value to the temperature-dependence of the underlying biochemical kinetics (e.g., *Johnson & Lewin, 1946*; *Sharpe & DeMichele, 1977*; *Schoolfield, Sharpe & Magnuson, 1981*; *Ikemoto, 2005*; *Corkrey et al., 2012*; *Hobbs et al., 2013*; *DeLong et al., 2017*) are becoming increasingly popular for quantifying empirically observed TPCs (*Hochachka & Somero, 2002*). Such models have occasionally received criticism on the grounds that they only constitute phenomenological statistical descriptions, as their assumptions are too simplistic and cannot be directly mapped onto physiological or ecological rates, which should be driven by a far more complex interplay of processes (e.g., *Clarke, 2004*; *Clarke & Fraser, 2004*; *Clarke, 2006*; *Clarke, 2017*; but see *Gillooly et al., 2006*). Nevertheless, these models continue to be used in the literature as they can adequately fit a large variety of experimentally determined TPCs, enabling the quantification of various aspects of the shape of the performance curve.

Among these models, the Sharpe-Schoolfield model (*Schoolfield, Sharpe & Magnuson, 1981*) has been frequently used in recent studies to address both ecological and evolutionary questions about the effects of temperature change on individuals, populations, and communities (*Barmak et al., 2014*; *Barneche et al., 2014*; *Fand et al., 2014*; *Simoy, Simoy & Canziani, 2015*; *Barneche et al., 2016*; *Padfield et al., 2016*; *Vimercati et al., 2016*). In particular, the $B_0$ calculated from fitting this model to TPC data has been used to compare the rate performance of different species (e.g., *Wohlfahrt et al., 1999*), treatments (e.g., *Padfield et al., 2016*), or developmental stages (e.g., *Hopp & Foley, 2001*) at a reference temperature, $T_{\text{ref}}$. However, the implicit assumption made by these studies, that $B_0$ is exactly the normalised rate value at $T_{\text{ref}}$, is only valid under certain conditions (see the Theoretical context section), and may in fact heavily overestimate the actual rate value at that temperature (*Schoolfield, Sharpe & Magnuson, 1981*) (Fig. 1). Such an overestimation could introduce unexpected biases not only in comparisons of temperature-normalised rates (among e.g., species, treatments, or developmental stages), but also in other analyses (e.g., exaggerating the rate performance of cold-adapted species could provide false support for MCA in its absence).

Here, we study the likely incidence of this overestimation of the normalised $B_0$ obtained by fitting the Sharpe-Schoolfield model to data of biological rates measured at a range of temperatures. To this end, we investigate the conditions under which this overestimation becomes particularly pronounced by analysing 1,758 real thermal performance curves across diverse ectotherm species and rates. We then show how conclusions based upon biased $B_0$ estimates can compromise the results of an important application of TPC models—detecting metabolic cold adaptation. Finally, we present alternative methods for obtaining realistic estimates of rate performance at a reference temperature under different scenarios of usage of the model.

## METHODS

### Theoretical context

The Sharpe-Schoolfield model proposes that the effect of temperature on the performance of a biological rate largely reflects the thermal sensitivity of a single rate-limiting enzyme that becomes deactivated at both extreme-high and low temperatures (*Schoolfield, Sharpe & Magnuson, 1981*). Nevertheless, low temperature inactivation is hard to detect, possibly because it requires multiple rate measurements at low temperatures for inferring accurate parameter estimates (see *Pawar et al., 2016*). Such resolution is typically lacking in currently available datasets of thermal performance. For this reason, it is usually more parsimonious to use a simpler version of the full model that ignores low-temperature enzyme inactivation (Fig. 1):

$$B(T) = B_0 \cdot \frac{e^{\frac{-E}{k} \cdot \left( \frac{1}{T} - \frac{1}{T_{\text{ref}}} \right)}}{1 + e^{\frac{E_D}{k} \cdot \left( \frac{1}{T_h} - \frac{1}{T} \right)}}. \tag{1}$$

Here, $B$ is the value of the rate at a given temperature $T$ (K), $E$ is the activation energy (eV), which controls the rise of the curve up to the peak, $E_D$ is the de-activation energy (eV), which sets the rate at which the rate falls after the peak, $T_h$ (K) is the temperature at which 50% of the enzyme units are inactive, and $k$ is the Boltzmann constant ($8.617 \cdot 10^{-5}$ eV K$^{-1}$). $B_0$ is the value of the rate at a reference (normalisation) temperature $T_{\text{ref}}$—i.e., $B_0 \approx B(T_{\text{ref}})$—assuming enzyme units are fully operational at that temperature. The model can also be reformulated without normalisation, but then $B_0$ would lose any biological meaning (see Section A2.1 in Appendix S1). The assumption of this model variant is that, at low temperatures, the population of the key enzyme remains fully active, with low rate performance values being driven by the decreased amount of kinetic energy which causes biochemical reactions to proceed at a very low rate.

*Schoolfield, Sharpe & Magnuson (1981)* originally suggested using $T_{\text{ref}} = 25$ °C, a choice they considered appropriate for most poikilotherm species. This suggestion has frequently been followed (see Table A1 and Fig. A1 in Appendix S1). However, when non-negligible loss of enzyme activity occurs at $T_{\text{ref}}$—e.g., due to denaturation or inactivation of some other component of the metabolic pathway— $B_0$ overestimates the real value of the rate at that temperature ($B(T_{\text{ref}})$) (*Ikemoto, 2005*). This is particularly problematic for comparisons of $B_0$ across diverse species, as significant temperature-mediated inactivation may begin at very different temperatures, potentially leading to different degrees of inaccuracy in the $B_0$ estimates.

### The inflation of rate value at reference temperature ($B_0$)

We first consider why $B_0$ can be biased. For this, in addition to the parameters in Eq. (1) ($B_0$, $E$, $E_D$, $T_h$, $T_{\text{ref}}$), two extra parameters need to be defined to capture all aspects of the shape of the TPC: the temperature at which the TPC peaks ($T_{\text{pk}}$), and the performance at that peak ($P_{\text{pk}}$; see sections A2.2-3 in Appendix S1 for their derivations). Setting $T = T_{\text{ref}}$ in Eq. (1) shows that the amount by which $B_0$ will deviate from $B(T_{\text{ref}})$ is equal to the

denominator of Eq. (1):

$$B(T_{\text{ref}}) = B_0 \cdot \frac{1}{1 + e^{\frac{E_D}{k} \cdot \left( \frac{1}{T_h} - \frac{1}{T_{\text{ref}}} \right)}}. \tag{2}$$

When $T_{\text{ref}}$ is much lower than $T_h$ (the temperature at which 50% of the enzyme units become inactive), $B_0 \approx B(T_{\text{ref}})$ because the denominator $\approx 1$. On the other hand, as the chosen $T_{\text{ref}}$ approaches $T_h$—or exceeds it—, $B_0$ will increasingly deviate from $B(T_{\text{ref}})$. In any case, $B_0$ will always be greater than $B(T_{\text{ref}})$ (at best, by a negligible amount) because of the denominator of Eq. (2). To explore this behaviour numerically across real TPCs of a single biological rate (for consistency reasons), we compiled a dataset of phytoplankton growth rates versus temperature (a combination of the *López-Urrutia et al., 2006*; *Rose & Caron, 2007*; *Bissinger et al., 2008*; *Thomas et al., 2012* datasets), containing 672 species/strains with growth rate being measured at multiple temperatures per species/strain. To each TPC in this dataset, we fitted the Sharpe-Schoolfield model across a range of $T_{\text{ref}}$ values ($-10$ °C to 30 °C) using the nonlinear least-squares method (Levenberg–Marquardt algorithm). In order to eliminate less reliable fitted parameter estimates, we rejected fits with (i) an $R^2$ below 0.5 (raising this cutoff to 0.9 yielded qualitatively identical results) or (ii) fewer than four data points either before or after $T_{\text{pk}}$. Based on these criteria, the number of accepted fits per $T_{\text{ref}}$ value ranged from 121 to 126 out of 672 starting TPCs (for an $R^2$ cutoff of 0.5). The variation in the number of retained parameter estimates is due to the different $T_{\text{ref}}$ values that we used which can cause small changes in the quality of the fit, leading to the occasional exclusion of some fits with $R^2$ values very close to the cutoff. The computer code—along with the names and versions of all modules or packages used—for the main analyses of this study (including fitting the Sharpe-Schoolfield model to TPCs) is available at https: //github.com/dgkontopoulos/Kontopoulos_et_al_temperature_normalisation_2017.

## Identification of conditions that lead to a severely overestimated $B_0$

We next determine the characteristics of TPCs (parameter combinations of the Sharpe-Schoolfield model) that lead to a severely overestimated $B_0$. This is a complex problem and not just a matter of determining the difference between $T_h$ and $T_{\text{ref}}$, because the denominator of Eq. (2) also includes the $E_D$ parameter. As $E_D$ influences the relationship between $T_h$ and $T_{\text{pk}}$ (see section A2.2 in Appendix S1), it is necessary to take into account the interplay of $T_h$ and $T_{\text{ref}}$ with $T_{\text{pk}}$. To address this, we use a conditional inference tree (a machine learning algorithm; *Hothorn, Hornik & Zeileis, 2006*) to determine the TPC model's parameter combinations that lead to strong overestimation.

For maximising the power of the machine learning method we used a larger dataset—a subset of the Biotraits database (a substantial collection of performance measurements of ecological traits and physiological rates at multiple temperatures from a wide range of species; *Dell, Pawar & Savage, 2013*) combined with additional data extracted from the published literature (see section A5 in Appendix S1). We first fitted the Sharpe-Schoolfield model to each empirical TPC in this dataset. As the dataset is very diverse—including, among others, rates from bacteria, macroalgae, and terrestrial plants—we set $T_{\text{ref}}$ to 0 °C so that we could obtain reasonable estimates (i.e., at a temperature below $T_{\text{pk}}$) of $B_0$

and $B(T_{ref})$ even for cold-adapted species with low $T_{pk}$ values. It is worth stressing that such a low $T_{ref}$ value is indeed appropriate because, as mentioned in the "Theoretical context" section, experimentally determined TPCs generally do not possess the required resolution for detecting low-temperature enzyme inactivation. Thus, it is safe to assume that rate estimates will be reasonable at low temperatures, even at 0 °C. In total, 1,758 species/individual curves were produced from this dataset. We did not filter the results based on goodness of fit metrics because we are interested in all the different parameter combinations regardless of how well they describe the data.

We then analysed this ensemble of fitted curves through the construction of a conditional inference tree from the data (see section A3.1 in Appendix S1 for details). More precisely, we specified a binary response variable: $B_0$ is above or below $P_{pk}$. The choice of $P_{pk}$ as the cutoff was due to the very high classification performance of the resulting model, especially when compared to other possible cutoffs (e.g., a three-fold increase from $B(T_{ref})$) which performed poorly. The predictor variables were the differences between (i) $T_{pk}$ and $T_h$, (ii) $T_{pk}$ and $T_{ref}$, and (iii) $T_h$ and $T_{ref}$ for each fit. The model was constrained by setting the maximum allowed $p$-value at each internal node below $10^{-10}$. Its performance was evaluated with the Matthews correlation coefficient (MCC; *Matthews, 1975*), a metric often used for machine learning models with a binary response. This metric takes values from $-1$ (complete disagreement with data) to 1 (complete agreement with data) and is considered reliable even when the different response states of the model (in this case $B_0 > P_{pk}$ and $B_0 < P_{pk}$) are not evenly sampled. To further ensure that the model was accurate and generalisable, we also estimated its performance against a distinct dataset of 405 TPCs (testing dataset). The data for these curves were also part of the Biotraits database—similarly to the 1,758 curves—but were not used for training the model.

**Implications of the inflation for investigations of thermal adaptation**

Among other ecological and evolutionary questions, the effects of adaptation to different thermal environments on the shape of the TPC (e.g., see *Huey & Kingsolver, 1989*; *Angilletta et al., 2003*; *Angilletta, 2009*; *Angilletta, Huey & Frazier, 2010*; *Clarke, 2017*) can be investigated using estimates from the Sharpe-Schoolfield model. For example, a study may aim to uncover whether there are any trade-offs between performance at lower and higher temperatures by correlating $B_0$ and $T_{pk}$ (e.g., a negative correlation would suggest that high performance at warmer temperatures would come at the cost of lower performance at colder temperatures). Overestimating $B_0$—especially for cold-adapted species with a $T_h$ value close to $T_{ref}$—may potentially introduce such correlations where none existed, serving as false-positive evidence for the MCA hypothesis.

To explore this possible issue, we generated a synthetic dataset of 1,000 negatively skewed TPCs, in which MCA was absent. While a real-world dataset of a single rate could also be used for this purpose (e.g., the phytoplankton growth rates dataset in Fig. 2), we resorted to a simulation in order to obtain a bigger sample and, more importantly, to ensure that the input data were not the outcome of the process of MCA. To this end, each curve was obtained by sampling from a distinct realisation of the beta distribution, with shape parameters ($\alpha$ and $\beta$; see section A4 in Appendix S1) that were in turn sampled

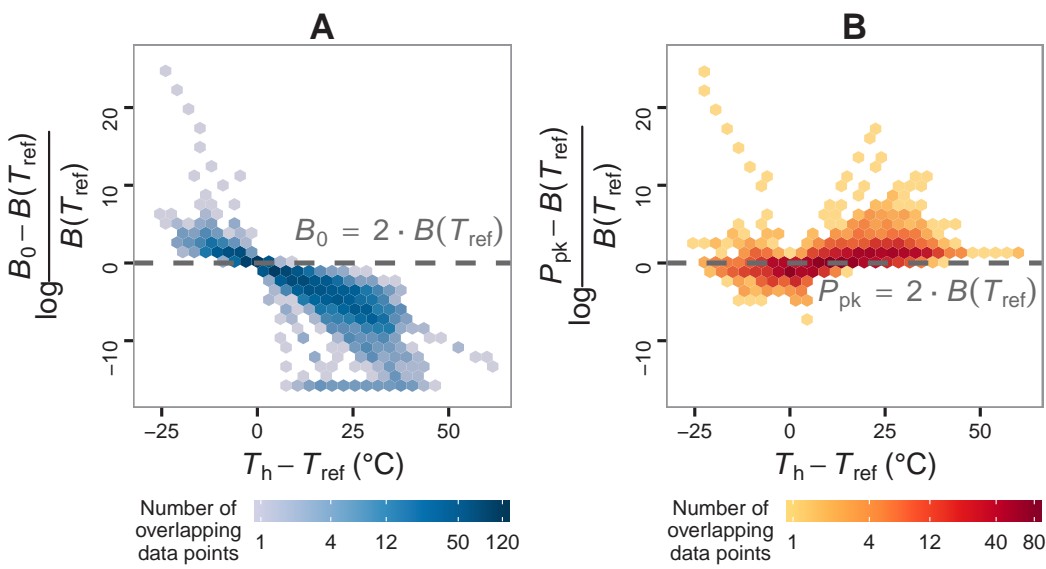

**Figure 2** **The effect of choice of reference temperature $T_{ref}$ on the deviation of $B_0$ from $B(T_{ref})$ (A) and its relationship with $P_{pk}$ (B).** The vertical axis of (A) stands for the log-fold increase of $B_0$ from $B(T_{ref})$, where a value of zero indicates that $B_0$ is double the real $B(T_{ref})$ value. Zero is used here as a reference point around and above which $B_0$ becomes non-negligibly exaggerated. Data points were obtained by fitting the Sharpe-Schoolfield model to a dataset of phytoplankton growth rate measurements versus temperature (see main text) across a range of $T_{ref}$ values. The colour depth of each hexagon is proportional to the number of data points at that location in the graph. As expected from Eq. (2), the deviation of $B_0$ from $B(T_{ref})$ decreases nonlinearly with the difference between $T_h$ and $T_{ref}$, to the point where the former asymptotically approaches zero (in linear scale). Towards the left end of the horizontal axis, the values of the estimates of $B_0$ even exceed those of the rate value at or close to optimum, $P_{pk}$.

from normal distributions (Table 1). Skewness was assessed by examining the $\alpha$ and $\beta$ parameters of each simulated curve. Curves that were not negatively skewed (i.e., those where $\alpha \leq \beta$) were removed and new ones were produced in their place. We also randomly varied the width and the height of the curves using two normally distributed parameters $j$ and $k$. As the minimum $T_{pk}$ in this simulation was at 8.23 °C, we arbitrarily set $T_{ref}$ to 7 °C, but any other $T_{ref}$ value below 8.23 °C could be used as well. Note that a different run of the simulation would most likely lead to a different minimum $T_{pk}$ value, which would potentially require a change in the chosen value of $T_{ref}$. To enforce the absence of MCA, we made sure that, in this population of curves, there was no significant association between the performance at a $T_{ref}$ of 7 °C, and the thermal optimum ($r = -0.03$, 95% CI [$-0.09$ to 0.03], $p = 0.35$).

We then fitted the Sharpe-Schoolfield model to each synthetic curve and obtained parameter estimates where possible. Following this, we performed two different tests for MCA, and compared the results when using $B_0$ versus $B(T_{ref})$. For the first test, the estimates were split onto two groups: (i) those originating from curves with $T_{pk} < 15$ °C (colder-adapted species), and (ii) those with $T_{pk} \geq 15$ °C (species adapted to warmer temperatures). We next tested whether the distributions of the normalised rates ($B_0$ and $B(T_{ref})$) were significantly different using the two-sample Kolmogorov–Smirnov test

Table 1 **Parameters for the generation of simulated curves.** $\alpha$ and $\beta$ are shape parameters of the beta distribution, whereas the two other parameters generate variation in the width and the height of the curves. $\beta$ is constrained to be smaller than $\alpha$, in order for the resulting curves to be negatively skewed, similarly to the observed thermal response curves of biological rates.

| Parameter name | Estimation |
| --- | --- |
| $\alpha$ | $\alpha \sim \mathcal{N}(\mu = 10, \sigma = 3)$ |
| $\beta$ | $\alpha - i, i \sim \mathcal{N}(\mu = 4, \sigma = 2)$ |
| Final curve width | original width $\cdot j, j \sim \mathcal{N}(\mu = 25, \sigma = 4)$ |
| Final curve height | original height $+ k, k \sim \mathcal{N}(\mu = 3, \sigma = 0.8)$ |

(*Corder & Foreman, 2014*). The second test consisted of a simple correlation between the normalised rate values ($B_0$ and $B(T_{ref})$) and the corresponding $T_{pk}$ values.

## RESULTS

### Conditions that lead to different degrees of inflation of $B_0$ estimates

Using the phytoplankton growth rates dataset, we show that, contingent on the difference between $T_h$ and $T_{ref}$, $B_0$ can be considerably greater than $B(T_{ref})$ (Fig. 2). More precisely, the deviation of $B_0$ from $B(T_{ref})$ decreases nonlinearly with the difference between $T_{ref}$ and $T_h$ (A). In many circumstances, the deviation of $B_0$ is extreme, becoming even greater than the rate value at or near optimum temperature, $P_{pk}$ (B).

The search for thermal response parameter combinations that lead to $B_0$ being above $P_{pk}$ (highly overestimated) or below it (less overestimated) resulted in a conditional inference tree with four terminal nodes (Fig. 3). In each of those nodes, $B_0$ was nearly exclusively below or above $P_{pk}$. This machine learning model exhibited high performance both on the training dataset (MCC = 0.954) and the testing dataset (MCC = 0.824; section A3.2 in Appendix S1). The sets of thermal response parameters in which $B_0$ was greater than $P_{pk}$ almost always had either a $T_h - T_{ref}$ difference that was less than 0.6 (relatively narrow curves), or a $T_{pk} - T_{ref}$ difference of 49.1 or lower (relatively wide curves).

### Impacts of the overestimation of $B_0$ on tests for MCA

In total, we were able to obtain thermal response parameter estimates for 968 simulated curves, as the nonlinear least-squares algorithm failed to converge on solutions for the remaining 32. In the first test for MCA the distributions of $B_0$ estimates differed between the two groups ($D = 0.18$, $p = 1.7 \cdot 10^{-6}$), with species adapted to colder temperatures having a higher median value of $B_0$ (Fig. 4A, light blue violin plots). In contrast, the two distributions of $B(T_{ref})$ estimates were statistically indistinguishable ($D = 0.07$, $p = 0.21$), as expected (Fig. 4A, green violin plots). The overestimation of $B_0$ also affected the second MCA test, as a weak negative correlation between $B_0$ and $T_{pk}$ was detected, but not between $B(T_{ref})$ and $T_{pk}$ (Figs. 4B and 4C). These results indicate that the inflation of $B_0$ can provide false support for the MCA hypothesis, even for datasets with complete absence of this pattern.

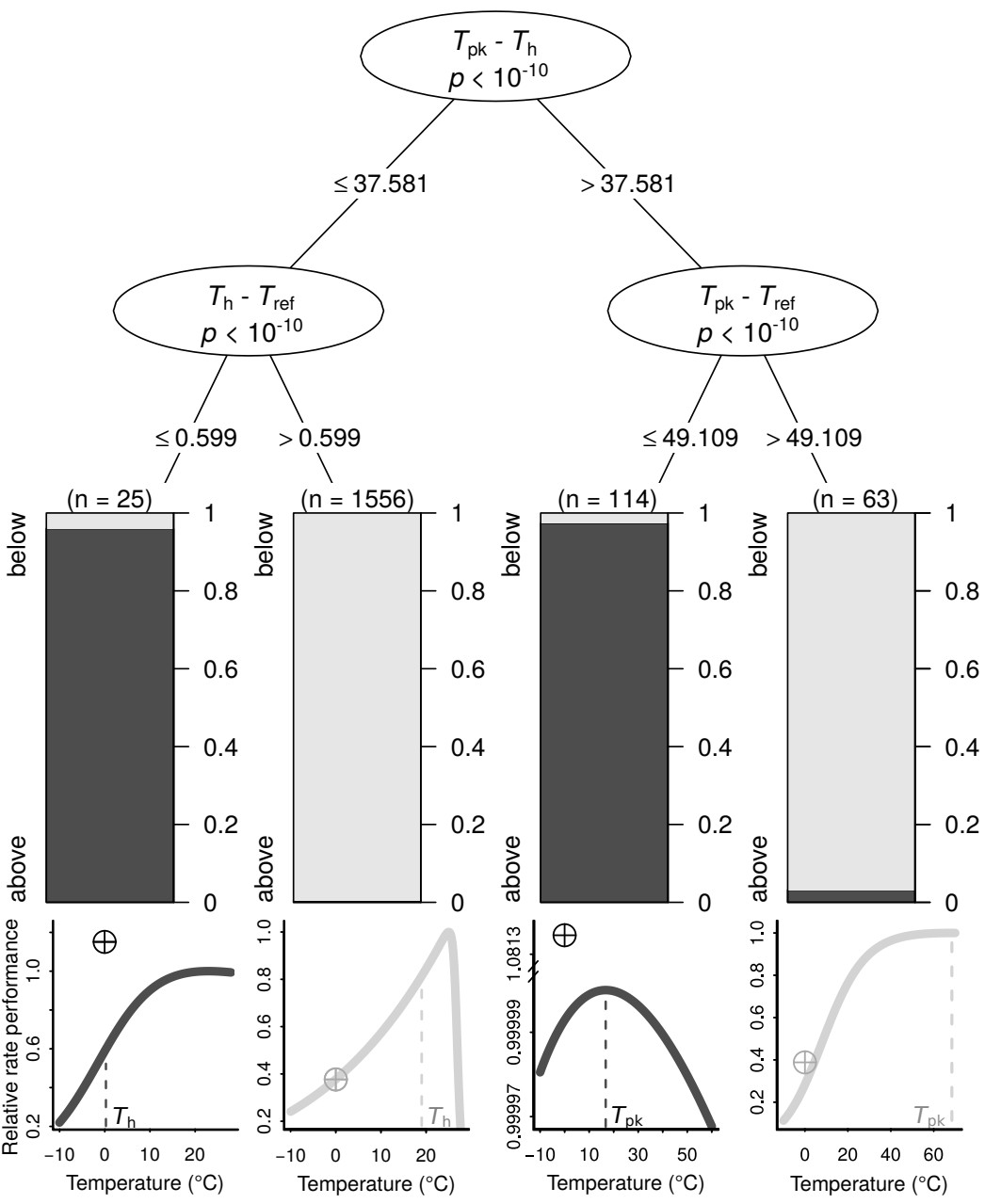

**Figure 3** **The conditions under which $B_0$ is highly overestimated (i.e., $B_0 > P_{pk}$; dark grey bars and curves) or less so (i.e., $B_0 < P_{pk}$; light grey bars and curves), determined using a conditional inference tree algorithm.** Representative examples of thermal performance curves, along with their $B_0$ estimates (crossed circles; normalised at 0 °C for consistency), are shown under each terminal node. The curves are not drawn on the same axes, as their rate performance values vary considerably, even if normalised relatively to the $P_{pk}$ value of each TPC. For a few very wide—and possibly biologically unrealistic—curves (right half), the difference between $T_{pk}$ and $T_{ref}$ determines whether $B_0 > P_{pk}$. In contrast, for the remaining curves, a $T_h$ value that is greater than $T_{ref}$ by more than 0.599 °C will always lead to $B_0$ estimates that are below $P_{pk}$.

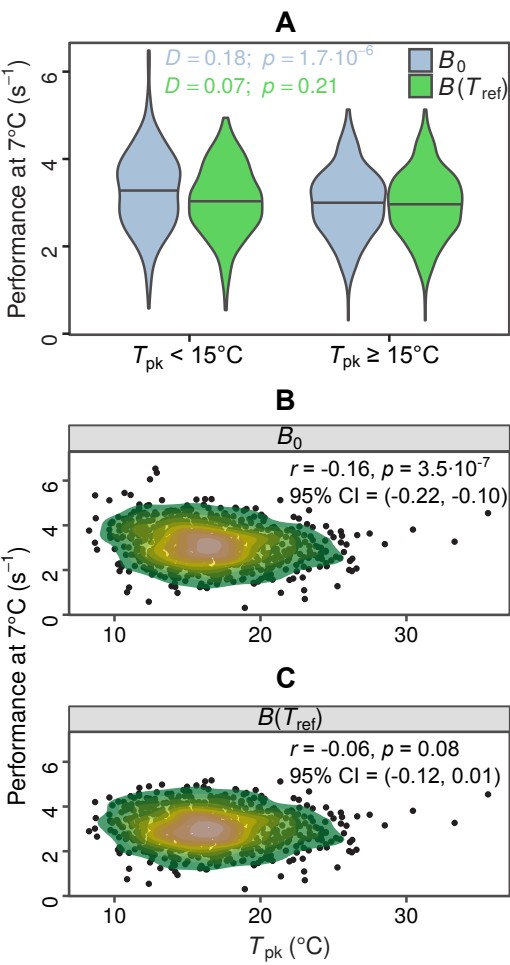

**Figure 4  Impacts of exaggerated $B_0$ estimates on tests for metabolic cold adaptation.** (A) Violin plots of rate performance at $T_{ref} = 7\,°C$, as estimated using $B_0$ (light blue) and $B(T_{ref})$ (green), for hypothetical cold-adapted species ($T_{pk} < 15\,°C$; left half) and species adapted to higher temperatures (right half). Horizontal lines indicate the median of each distribution. The statistical significance of the difference in performance between the two temperature groups was evaluated according to the two-sample Kolmogorov–Smirnov test. Based purely on the $B_0$ estimates—which get increasingly inflated at low temperatures as $T_h$ approaches $T_{ref}$—one would mistakenly conclude that metabolic cold adaptation is present in this dataset. (B, C): Correlations of $B_0$ with $T_{pk}$, and $B(T_{ref})$ with $T_{pk}$. The color surfaces represent the local density of data points. A similar pattern to the previous panel emerges, as the inflated $B_0$ estimates—in contrast to the true values—suggest that cold adaptation is present, albeit weakly.

## DISCUSSION

In this paper we have addressed the consequences of estimating the value of a rate at a reference temperature, $B_0$, using the Sharpe-Schoolfield model, but without satisfying one of its fundamental assumptions: that the key enzyme—which is responsible for the temperature dependence of the rate—is fully functional at the reference temperature. When this assumption is not met, $B_0$ will overestimate the real rate performance at the reference temperature, $B(T_{ref})$ (*Ikemoto, 2005*).

We explain how the discrepancy between $B_0$ and $B(T_{\mathrm{ref}})$ arises and determine the conditions under which it becomes particularly pronounced using a machine learning approach (Fig. 3). The resulting conditional inference tree shows that $B_0$ estimates will generally exceed the rate performance at the peak of the curve ($P_{\mathrm{pk}}$) as long as: (i) $T_{\mathrm{pk}} - T_{\mathrm{h}}$ is less than ~37.58 °C and $T_{\mathrm{h}} - T_{\mathrm{ref}}$ is less than ~0.6 °C, or (ii) $T_{\mathrm{pk}} - T_{\mathrm{h}}$ is greater than ~37.58 °C and $T_{\mathrm{pk}} - T_{\mathrm{ref}}$ is less than ~49.11 °C. In any other case, $B_0$ would most likely be smaller than $P_{\mathrm{pk}}$, although its inflation may well still be of concern. Using a synthetic dataset, we then demonstrate that wrongly assuming $B_0 = B(T_{\mathrm{ref}})$ can lead to erroneous conclusions in analyses of thermal adaptation, as the overestimation of $B_0$ can mimic the effects of metabolic cold adaptation (Fig. 4) (a Type I error).

It is important to note that while we focus on the four-parameter version of the Sharpe-Schoolfield model in this study, the inflation of $B_0$ estimates also mathematically occurs in the variant of the model that assumes enzyme inactivation at both high and low temperatures. Thus, caution is warranted regardless of the model variant that is chosen. Beyond this issue, fitting the simpler model instead of its full counterpart may potentially give rise to other inherent biases but, to our knowledge, a thorough comparison of the two model variants across different organismal groups and rates is not available.

As mentioned before, previous studies have tended to set the $T_{\mathrm{ref}}$—usually at a value of 25 °C—while fitting the Sharpe-Schoolfield model without considering the potential inflation of $B_0$ (Table A1 and Fig. A1, Appendix S1). Whether results of these studies have been compromised by an inappropriate use of $T_{\mathrm{ref}}$ is impossible to determine definitively because most of these studies report either $T_{\mathrm{h}}$ or $T_{\mathrm{pk}}$ estimates, whereas the machine learning model depends on both (see the 'Conditions leading to a severely overestimated $B_0$' section), along with the value of $T_{\mathrm{ref}}$. If these data were available, using the machine learning model that we generated would provide a straightforward procedure to identify cases where $B_0$ is highly likely to be extremely overestimated (i.e, greater than $P_{\mathrm{pk}}$). In fact, the only study where all necessary parameter estimates were reported for all fitted curves was that by *Padfield et al. (2016)*. In that study, the maximum difference of $T_{\mathrm{h}}$ from $T_{\mathrm{pk}}$ is 2.49 °C, and the minimum difference of $T_{\mathrm{ref}}$ from $T_{\mathrm{h}}$ is 5.79 °C, which, according to the machine learning model (see Fig. 3), are sufficient for the $B_0$ estimates to be below those of $P_{\mathrm{pk}}$. Having said that, as we showed in this paper, the fact that the overestimation of $B_0$ is not extreme does not necessarily rid any drawn conclusions of bias (e.g., the possibility of falsely detecting the effect of MCA).

In any case, it is crucial to point out that choosing an appropriate reference temperature (i.e., one that is low enough but within the temperature range that the species can endure) is not—on its own—a sufficient strategy to avoid the overestimation of $B_0$. As different species or individuals will most likely not share a common $T_{\mathrm{h}}$ value, the difference between $T_{\mathrm{h}}$ and $T_{\mathrm{ref}}$ will vary across the dataset (see Fig. 2). This approach could again lead to an exaggeration (which may however be very small) of some $B_0$ estimates and is therefore not an elegant solution to the problem.

## Comparisons of temperature-normalised rates of diverse species
### When data span the entire TPC

For studies in which the end goal is to compare the performance of different species at a common temperature, the simplest approach would be to fit the Sharpe-Schoolfield model—with or without normalising $B_0$ at a reference temperature—and compare estimates of $B(T_{ref})$, calculated a posteriori. The confidence intervals around $B(T_{ref})$ can then be estimated by bootstrapping. Another option to avoid the issue of rate overestimation is to consider fitting other models, such as the macromolecular rates model (*Hobbs et al., 2013*) or the enzyme-assisted Arrhenius model (*DeLong et al., 2017*).

### When data only cover the rising part of the TPC

While the previous solutions are applicable to thermal response datasets that capture either the rise of the curve or its entirety, few studies report temperature performance measurements after the unimodal peak of the response (*Dell, Pawar & Savage, 2011*). Therefore, to obtain an estimate of baseline performance from a dataset that only covers the exponential rise component, one could instead fit the Boltzmann-Arrhenius model (e.g., see *Gillooly et al., 2001*),

$$B(T) = B_0 \cdot e^{\frac{-E}{k} \cdot \left(\frac{1}{T} - \frac{1}{T_{ref}}\right)}, \tag{3}$$

which does not suffer from the problems of the Sharpe-Schoolfield model, as $B(T_{ref})$ indeed simplifies to $B_0$.

A second alternative model is the one that includes the $Q_{10}$ factor (see *Gillooly et al., 2001*), i.e., the rate of change in biological rate performance after a temperature rise of 10 °C:

$$Q_{10} = \left(\frac{B(T_2)}{B(T_1)}\right)^{\frac{10}{T_2 - T_1}}. \tag{4}$$

In this case, one would first estimate the value of $Q_{10}$ from known rate values at two temperatures, and use it to calculate the rate value at the reference temperature:

$$B(T_{ref}) = B(T_1) \cdot Q_{10}^{\frac{T_{ref} - T_1}{10}}. \tag{5}$$

Regardless of which of these two models is chosen, careful attention must be paid to ensure that the biological rate increases exponentially across the entire temperature range, without signs of a plateau being reached. Otherwise, the estimates may yet again be biased.

## Using the 'intrinsic optimum temperature' instead of $T_{ref}$

Alternatively, baseline performance could be defined as the height of the curve at the temperature where the population of the key enzyme is fully active, which should be characteristic for each individual or species. In the Sharpe-Schoolfield model, the denominator indicates the percentage of enzymes that are active. Therefore, in the four-parameter variant of the model, the intrinsic optimum temperature could be estimated as the highest temperature at which this percentage is sufficiently high (e.g., at 99%). If, instead, the model of choice is the Sharpe-Schoolfield variant that also accounts for

enzyme inactivation at low temperatures, there will be a unique temperature at which the enzyme population is 100% active. Otherwise, the intrinsic optimum temperature can also be obtained from the Sharpe-Schoolfield-Ikemoto (SSI) model (*Ikemoto, 2005*). This model integrates the law of total effective temperature—often used in studies of arthropod or parasite development—within the Sharpe-Schoolfield model, replacing $T_{\text{ref}}$ with the intrinsic optimum temperature. However, this model introduces an extra parameter and is more challenging to fit compared to the original Sharpe-Schoolfield model. To mitigate this problem, software implementations have been developed that reduce the computation time from often more than 3 hours (*Ikemoto, 2008*) down to less than a second (*Shi et al., 2011*; *Ikemoto, Kurahashi & Shi, 2013*).

## CONCLUSIONS

Obtaining accurate estimates of temperature-normalised rate performance is of crucial importance—especially in the face of climate change—for comparisons of the same rate across different organisms, or different rates within an individual. In this context, our study explains why temperature-normalised rate estimates obtained using the Sharpe-Schoolfield model can be strongly exaggerated—in comparison to the true rate values—when one of the assumptions of the model is violated, and gives an example of possible consequences of this exaggeration. The suggestions that we provide to address this issue should be useful to the burgeoning studies on ectotherm thermal performance and climate change, both for performing meta-analyses and for determining appropriate temperature ranges in laboratory experiments.

## ACKNOWLEDGEMENTS

We thank Timothy G. Barraclough, James Rosindell, Jonathan Lloyd, and Gabriel Yvon-Durocher for useful discussions. We also thank two anonymous reviewers for their insightful and thorough comments. Finally, we are grateful to Panagiota S. Georgoulia for providing the atomic coordinates of native and partly denatured protein structures that were shown in Fig. 1. This paper is devoted to the memory of Terpsithea Karaggelou - Kontopoulou.

### Funding

Dimitrios - Georgios Kontopoulos was supported by a Natural Environment Research Council (NERC) Doctoral Training Partnership (DTP) scholarship (NE/L002515/1). Thomas P. Smith was supported by a Biotechnology and Biological Sciences Research Council (BBSRC) DTP scholarship (BB/J014575/1). Bernardo García-Carreras, Sofía Sal, and Samraat Pawar were supported by a NERC grant awarded to Samraat Pawar (NE/M004740/1). The funders had no role in study design, data collection and analysis, decision to publish, or preparation of the manuscript.

## Grant Disclosures

The following grant information was disclosed by the authors:
Natural Environment Research Council (NERC).
Doctoral Training Partnership (DTP) scholarship: NE/L002515/1.
Biotechnology and Biological Sciences Research Council (BBSRC).
DTP scholarship: BB/J014575/1.
NERC: NE/M004740/1.

## Competing Interests

The authors declare there are no competing interests.

## Author Contributions

- Dimitrios - Georgios Kontopoulos conceived and designed the experiments, performed the experiments, analyzed the data, wrote the paper, prepared figures and/or tables.
- Bernardo García-Carreras and Samraat Pawar conceived and designed the experiments, reviewed drafts of the paper.
- Sofía Sal and Thomas P. Smith reviewed drafts of the paper, provided some of the data that were used in this study.

## Data Availability

The raw data are included in the Supplemental Files, and the source code for the analyses of this study is available at https://github.com/dgkontopoulos/Kontopoulos_et_al_temperature_normalisation_2017.

## Supplemental Information

Supplemental information for this article can be found online at http://dx.doi.org/10.7717/peerj.4363#supplemental-information.

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
