# Peer review of "Use and misuse of temperature normalisation in meta-analyses of thermal responses of biological traits"

_PeerJ, doi:10.7717/peerj.4363_

## Round 0.1 · original submission · Major Revisions

The reviewers have made some excellent suggestions for improving your work. Please clearly respond to each comment, and carefully consider all points made in your response letter. In particular, Reviewer 2 raises some issues about the interpretation of enzyme versus whole organism level performance. I would like this issue to be clearly articulated with the appropriate caveats and I will send your revision out for a second review.

Reviewer 1 ·

Basic reporting

The article is generally well written in unambiguous language, sufficient introduction material and background with additional provision of theoretical context which greatly enhances readability for non-expert audience. In my opinion, the article would benefit from consistent use of the terms ‘traits’ or ‘rates’, which I have highlighted throughout in the PDF and ‘general comments’ section of the review. In additional some restructing in the introduction is suggested in the ‘considerations’ and additional points to consider which I think are essential for the discussion are also highlighted in the ‘considerations’ section of my peer review.

There is no statement of submission of data to a public archive, and the data in the supporting information did not download with correct formatting for me to adequately re-use or comment on suitability for re-use (perhaps a problem with a file conversion through the peer-review process).

Other than these comments, I have no major problems with the basic reporting of the article.

Experimental design

A clearly defined, and well tested, research question is explored throughout the manuscript which is undertaken to an impressive technical standard.

Lines 97-100 should clarify further the conditions for inclusion of these data in the article, additional a sentence describing ‘Biotraits’ on line 114 would be useful. Clarification is needed as to the software and packages used throughout the manuscript for full replication.

Validity of the findings

The general validity of these findings is likely to be high based on the robust analytical techniques and extensive database used, I believe the simulations are well designed an appropriately analysed and interpreted.

Additional comments

Thank you for providing an insightful and thought-provoking piece of work exploring the parameterisation of thermal performance curves and the consequent inflation of standardised performance estimates to review. It is clear the analyses are carefully orchestrated and undertaken with rigor and the resulting article is an interesting read. It is testament to the research that the article raises interesting questions which could be answered, I believe, within this piece if work – however I would consider these suggestions are at the authors discretion as the analyses are, in my opinion, sufficient in their current state. These suggestions are presented below with ‘considerations’ and ‘line-by-line’ points identified. I believe I have identified some necessary revisions to the manuscripts figures and additional points of discussion that should be addressed.

Considerations:
The article generally ignores discussing another key assumption that “a simpler version of the full model that ignores low-temperature enzyme inactivation is most often use” other than to acknowledge it. This assumption could be more strongly criticised than the assumption that Th is greater than Tref. While I don’t think it would be necessary for this particular publication that this assumption is explored (although it would be interesting and add to the paper), I think at least a discussion of the potential implications of this additional assumption should be included in the discussion, given that a seemingly more minor assumption of the model is explored in such detail throughout. Additionally, the analysis initially raised my concerns that a temperature of 0°C was chosen at Tref where I assume enzyme activity will be very low – while the assumption discussed here isn’t the main point of the article it would be better to acknowledge and discuss the implications of this assumption explicitly to avoid potential criticism that the analyses presented here ignores a potentially more important key assumption.

I light of the above comment, I think the analysis in “Identification of conditions that lead to a severely overestimated B0” would benefit from separating ‘cold adapted’ and ‘warm adapted’ species such that the assumption that enzymes are active at 0°C does not apply to the warm adapted species (i.e., set Tref to different values for these two groups? Or apply 0°C and 15°C Tref to both) – I think that the assumption that enzymes operate above 50% activity at 0°C for a tropical poikilothermic species is very weak, and the authors have a large dataset to choose to further test these ideas and assumptions.

The analyses using real thermal performance rate variation across temperature gradients may be strengthened by comparisons with simulations. For example, in the analysis “Inflation of trait value at reference temperature (B0)” a simple simulation coupled with the analysis of real data would provide an interesting comparison and be less biases by available data. Additionally, the behaviour of the simulation would be independent of biological heterogeneity. A comparison of expectations based on theory (simulation) with the resulting inflation of B0 in real traits would provide further insight to the magnitude of real inflation compared to potential for inflation (simulation) (i.e., is this as large a problem in real data as it could be?).

In the analysis “Implications of the inflation for investigations of thermal adaptation” the choice of Tref has been demonstrated as important in the previous analysis but is given as a fixed value here. Given that this is a simulation it would be relatively trivial to re-run over multiple values of Tref to explore the interaction between Tref and the negative relationship between Tpk and B(Tref) – I think this is important as currently I think the support for an ‘artificial’ relationship between Tpk and B(Tref), and thus false identification of MCA, is quite weak given the strength of the correlation. Furthermore, it would be better to justify where the choice of 7°C arises, other than it is below the minimum Tpk

A second suggestion for the analysis of “Implications of the inflation for investigations of thermal adaptation” that would support the take-home message of the paper, is to definitively show that MCA is not falsely detected when using the alternative approaches/models put forward by the authors. This should also be relatively simple given the simulation approach taken but would strengthen the authors argument that the Sharpe-Schoolfield model may falsely identify MCA but other types of models will not make the same error, and that therefore this is due to the assumption of the SS model.

Figure 2 isn’t as clear as it could be, although the information is fundamental to the paper. First, I think the equation defining the y-axis may be incorrectly printed as if B0 or Ppk is greater than B(Tref) then the equation suggests we take the log of a negative number? Second, I think the differences between B0 and B(Tref) should be presented in both absolute and proportional terms (relative to initial value and maximum rate – thus creating a 4-panel plot) because a doubling of a rate from 0.01 to 0.02 when the maximum value of the rate is 200 could be interpreted as less of a ‘problem’ compared to a lower proportional rate inflation of 10 to 15 which has a much larger absolute change. Second, the y-axis is very confusing and a value of ‘0’ to define when changes are non-negligible seems odd to me, it would be better if this were transformed such that data at 0 is a negligible change in B0 from B(Tref), and differences get larger with increasingly positive y-axis. Perhaps even a log(response ratio) may be useful here for proportional comparisons between B0 and B(Tref)?

On line 122-123 I’m not completely convinced that applying a cut off whereby rates = 0 at B0 were removed is valid, many species are likely to have performance rates at 0 for certain values of B0 depending on the thermal environment the species is adapted to. Can it be demonstrated this does not bias results, or that this cut-off was relevant for a small proportion of species (i.e., n=2).

Consistency in the use of ‘rates’ and ‘traits’ would be conceptually tighter. I personally would consider most TPCs to be derived from rates, rather than traits, and that the term ‘rates’ is more specific.

My supporting data files didn’t download properly, and a conversion from text to data in excel did not produce a dataset with standardised columns. Perhaps this is from transfer between the journal but may be worth double checking the files.

Line-by-line considerations:
4: This is a funding body/PhD program rather than an address?

26-32: I’m not convinced that the introduction benefits from this paragraph greatly. I think the work presented here is more broadly applicable to accurately understanding thermal variation in traits and comparative thermal physiology, which has relevance for climate change, but climate change shouldn’t lead the introduction in my opinion as your research is more fundamental and climate change isn’t relevant to later discussions.

24: Suggest changing keywords ‘model’ and ‘trait’ , ‘rate’ as very broad.

26-32: Given that the main importance the article highlights of accurately estimating B0 I would consider introducing metabolic cold adaptation early in the introduction

29: Garcia et al. 2014 isn’t relevant here.

30: Suggest changing ‘biological traits’ to ‘biological rates’

33: Suggest changing ‘are unimodal’ to ‘are generally unimodal’. Suggest changing ‘trait-value’ to ‘biological rate’

34: Suggest changing ‘data’ to ‘relationships’.

39-41: In my opinion, the introduction would benefit from a broader discussion of how the parameters are used in comparative physiology (not just MCA), with an emphasis that detection of MCA is an example of where inaccurate parameterisation of models can lead to false-positive results (I’m sure however there are many others too, but obviously MCA is useful to mention as the analysis investigates this effect later).

42: “the traits value” sounds odd. Which trait? ‘a cellular, organismal or population rate’s value’?

58: Could the ending of this sentence please be more explicit by defining ‘to data’. (e.g. to thermal performance rates).

42-56: In my opinion, this paragraph could benefit from a brief sentence describing the explicit consequences of overestimating B0 and how overestimation could alter current understanding of metabolic cold adaptation.

60: rates

79-82: Is this list of references is necessary given Table A1?

96-97: This statement on first reading appears contradictory to the result presented in figure 1A this is because the y axis on figure 1 is very non-intuitive and could be improved. Based on this statement, I initially thought why at the minimum difference between Th and Tref is the deviation of B0 from B(Tref) at a minimum too, if I misinterpreted this with I’m sure others will too. Consider rescaling such that the difference between B0 and B(Tref) is minimal at 0 on the y-axis.

104-105: Ensure the number of rejected vs. potential TPCs is recorded in the manuscript based on these cut-offs.

105: I would be interested to know that the results do not depend on the R2 cut off.

111-114: Please be specific about the machine learning approach used on first mention.

116-119: I do not follow the logic of using a Tref that = 0°C to obtain “reliable estimates of B0” when the argument put forward in the article is that “when non-negligible loss of enzyme activity occurs at Tref – e.g., due to denaturation or inactivation of some other component of the metabolic pathway – B0 overestimates the real value of the trait at that temperature (B(Tref))”. I may be wrong but I cannot imagine that >50% enzymes would be active at 0°C, I think this would be valid if the aim of this is to artificially inflate B0? It isn’t clear to me if this is the aim here. See ‘Considerations’.

122-123: I’m not completely convinced this cut off couldn’t bias your results. See ‘Considerations’.

124-136: In my opinion, the precise aim of this analysis should be introduced in more detail prior to describing the modelling process with a conditional inference tree. It is not clear in this paragraph what the ‘predictor’ variables are, and found it difficult to follow the paragraph and its relevance as a consequence.

134: ‘assess overfitting’

164: Your data are larger than your simulation of 1,000 TPCs, so this isn’t a valid justification (although I support the use of a simulation here).

164: consider rephrasing difficult to read.

167-168: How was ‘negative skew’ was assessed.

175-176: Is there a reference which justifies that species <15°C Tpk are ‘cold’ adapted?

201-203: The low value of the correlation coefficient should be emphasised here as it is in the legend of Figure 4.

210-211: This sentence would benefit from emphasising whether this is a novel finding of the study.

212-213: The first sentence here could be more clearly articulated. The word ‘this’ is used twice and thus the meaning of the sentence is lost, consider rephrasing. For example, “We determined that the discrepancy between real trait performance and estimated trait performance at B0 arises under the conditions that…”

216-217: This paragraph would benefit from some assessment as to whether most of the literature would suffer from this error, I would assume this is the case in a few studies of cold adapted species.

223-225: Is there a way to build the ML model such that the inappropriate use of Tref can be assessed by the parameters available in the literature?

232: ‘ones’ sounds informal.

233-234: Consider rephrasing as meaning is not initially clear to me. Is this sentence saying there may still be a small amount of bias in B0?

235: Unsure if this sentence is necessary?

238: A further useful suggestion could be to use a Tref appropriate based on the metabolic group such that a Tref of 25°C is not used for stenothermic polar species (for example).

261: This feels like it should be in the ‘recommendations’ section rather than a stand-alone header.

265-266: This sentence suggests there would only be a single temperature at which enzyme activity is at 100%, is this the case?

277-279: In my opinion, this sentence slightly overstates the implications of the results. I don’t think the patterns demonstrated in this analysis justify a ‘strong exaggeration’ of temperature-normalised trait-estimates and the correlation coefficient in figure 4 is quite low and, as such, the consequences of this rate estimate exaggeration appear to be relatively minimal.

284-285: Is there a NERC funding grant no. attached to each DTP?

Annotated reviews are not available for download in order to protect the identity of reviewers who chose to remain anonymous.

Reviewer 2 ·

Basic reporting

The basic reporting meets most of the criteria. However, they seem to be missing some key citations (see specific comments) from the literature.

Experimental design

The authors achieved their research goal and provided an answer to their question. However, they left out a few key talking points (see specific comments).

Validity of the findings

The authors demonstrated some problems with the Sharpe-Schoolfield model, but need to consider enzyme and whole-organism performance separately. Thus, I don't find the results particularly interesting and even a little misleading at times.

Additional comments

General comments:

Here the authors evaluate the use of the Sharpe-Schoolfield equation to normalize data for making comparisons of organismal performance. Overall, the authors make a sound argument that the Sharpe-Schoolfield model generates biased estimates of temperature normalized performance when key assumptions are violated. However, the results are underwhelming and I have reservations as to why anyone would use such a model for whole-organism or population data. See papers (below) by Andy Clarke which explain why such models should never be confused with a description of the mechanisms underlying performance, which are undoubtedly fare more complex than a rate-limiting enzyme or even enzymes alone. Thus, the model’s parameters can only be interpreted at the enzyme level, rather than at the organismal level. At the very least the authors should address this issue in their discussion. Furthermore, the authors discuss and seem to promote the use of the Arrhenius equation as an alternative method. However, Andy Clarke (2004; 2006) provides good arguments against molecular interpretations of the Arrhenius equation when fit to the metabolic rates of organisms. This leads me to question the relevance of the study in general. Organismal biologists will likely not find this paper very useful as one should not draw conclusions about enzyme performance from whole-organism performance.

Perhaps a more specialized journal (e.g., comparative and evolutionary biochemistry) would be more appropriate than PeerJ.

Clarke, A. (2004) Is there a Universal Temperature Dependence of metabolism? Functional Ecology, 18, 252-256.

Clarke, A. (2006) Temperature and the metabolic theory of ecology. Functional Ecology, 20, 405-412.



Specific comments:

Lines 35-37: This is an unconventional way of describing a thermal performance curve and I’m not aware of any ecologists that actually quantify these four values. The only citation provided for use of these values is a self-citation, which makes me even more skeptical. For example, optimal temperature (To) is the common term here, not “the temperature where the performance peaks (Tpk)”. I don’t much care for acronyms, and trying to introduce a new version of an existing acronym is unnecessary and confusing to the field. Ecologists are also concerned with the more common terms “performance breadth” and “critical thermal limits” (e.g., CTmin and CTmax).

Lines 218-220: It seems odd that this point (a single sentence) needs to be its own paragraph. Either elaborate on the point or work it into another paragraph.

Lines 242-244: Why should these other models be considered? Provide a sentence or two describing the validity of their application here.

Lines 247-249: I don’t agree with this statement as numerous studies (almost all?) that measure performance curves measure performance below and above optimal temperatures. After all, that is how optimal temperature is determined. See:

Angilletta (2006) Estimating and comparing thermal performance curves.

Vickers et al. (2011) Extending the cost-benefit model of thermoregulation: high-temperature environments.

Williams et al. (2016) Biological impacts of thermal extremesL mechanisms and costs of functional responses matter.


Lines 261-273: This is the most common, and most appropriate, approach researchers use when studying whole-organism performance.

---

## Round 0.2 · Minor Revisions

The previous Editor for your submission is unavailable, so I have been asked to step in and take it over.

Thank you for your responses to the first round of reviews - the reviewers appreciated your thoughtful consideration. Please respond with equal care to the reviewer's remaining comments.

Reviewer 1 ·

Basic reporting

I thank the authors for including MCA in the introduction, but feel it's definition is too vague and little introductory information is provided on MCA. At least a specific definition of MCA would improve the introduction, particularly now that the paper is more focussed on this topic.

In my opinion the “Theoretical Context” section hangs awkwardly between the introduction and the methods, moving to the methods may improve the flow of the article.

The discussion does a good job in suggesting solutions to help compare temperature normalised rates, but would benefit from some specific examples of MCA 'false positives' based on incorrect parameterisation of TPCs, there is a broad literature, emphasised in Appendix A. The discussion would be more thorough and constructive if these references were used to place the results relating to MCA false detection in a richer context.

Experimental design

Including an equation of how the beta-distribution is parameterised would be necessary to understand how the ‘final curve height’ and ‘final curve width’ of the Table 1 are used. I cannot see how these are used in the parameterisation of the beta distribution. Subscripts i, j, and k, are not explained in the text or table 1.

Validity of the findings

No comment.

Additional comments

Thank you for your thoughtful and detailed responses to each of my comments. I think the updated manuscript is much improved and have only minor comments in the PeerJ review, and a few detailed points below.


Line by line considerations:
69: Reword to contain example in parenthesis within sentence for clarity, “aforementioned and other analyses” sounds vague.

134: Is the text for this subheading intentionally smaller?

140: unintentional double space

141: Unnecessary - “again using a large empirical dataset”

143: What is the total sample size of individual TPCs including training and test sets?

148: Would be beneficial to see a summary table of the taxonomic coverage/trait coverage in the SOM. Bacteria, macroalgae and terrestrial plants seems an odd choice of organismal diversity to choose to emphasise Biotraits scope.

169: Was this a subset of the Biotraits/literature. The sentence sounds like this was a separate dataset at the moment? Data subset?

184: Unnecessary

189: Please specify that the alpha and beta parameters are parameters of the Beta distribution.

200: Consider rephrasing this paragraph as I find it surprisingly difficult to understand despite the relative simplicity of the procedure.

209: The results start with a statement to direct the reader to a figure rather than a statement of the result with the relevant figure in parenthesis.

210: Could this be stated in terms of interpretation of the parameter differences rather than the parameters themselves? The reader currently has to juggle how the parameter equations relate to concepts and then to results in one go.

Figure 2. Could the Figure 2 axis y label be phrased in words rather than equations, with equations in brackets perhaps? (i.e., Deviation of B0 from B(Tref), Deviation of optimum performance from B(Tref)). I feel like this figures requires a lot of work from the reader and axis simplification would remove on component of the complexity.

Annotated reviews are not available for download in order to protect the identity of reviewers who chose to remain anonymous.

Reviewer 2 ·

Basic reporting

No new comments

Experimental design

No new comments

Validity of the findings

No new comments

Additional comments

I appreciate the authors’ responses to my comments and the addition of some additional references to strengthen their arguments. This is a much improved version of the manuscript.

The only concern I have at this point is the author’s interpretation of optimal temperature. I think they are oversimplifying the measure and actually just describing it with their new proposed acronym (point at which the performance curve peaks). I feel that this is unnecessary and confusing to new readers in the field of thermal biology, but shouldn’t prevent the paper from being published. Readers can decide to follow this new acronym or not.

Specifically, I’m not sure how the authors are interpreting “optimal temperature”, but my understanding is that it is the temperature at which “optimal performance” is experienced produces the “highest peak” of a performance curve for that given trait. They could use their modified definition to simply describe what optimal temperature (a much better known term) is. I agree that where some performances are optimal (i.e., highest peak) does not automatically mean the “organism’s fitness is maximal”. In fact, the only times that is possibly correct is when the performance curve is specifically measuring fitness or a given trait’s optimal temperature overlaps with the temperature that produces maximal fitness. However, this is rarely the cased as most performance curves are more specific than “fitness” (or measure different proxies for fitness) and measure a single trait such as sprint speed, digestion rate, or enzyme reaction rates.

Yes, organisms often experience temperatures below optimal temperatures as these temperatures are energetically costly and unnecessary at all times. For example, an animal has no need to be at optimal temperature for digestion when it has an empty stomach. It makes ecological sense to be below the optimal temperature for digestion when without food in order to conserve energy. Furthermore, this is why there are often different optimal temperatures (or performance peaks) for different traits (e.g., sprint speed vs assimilation rate).

And finally, according to a brief literature search through Google Scholar, “the temperature where the curve peaks” did not produce any direct hits whereas “optimal performance” produced hundreds if not thousands of hits.

---

## Round 0.3 · accepted · Accept

Thank you for your careful consideration of the reviews. I am happy to accept this version!